# Erbium-Doped GQD-Embedded Coffee-Ground-Derived Porous Biochar for Highly Efficient Asymmetric Supercapacitor

**DOI:** 10.3390/nano12111939

**Published:** 2022-06-06

**Authors:** Thi Ai Ngoc Bui, Trung Viet Huynh, Hai Linh Tran, Ruey-an Doong

**Affiliations:** 1Department of Biomedical Engineering and Environmental Sciences, National Tsing Hua University, 101, Sec. 2, Kuang Fu Road, Hsinchu 30013, Taiwan; buiaingoc.ep03g@g2.nctu.edu.tw (T.A.N.B.); tranlinhhai@gmail.com (H.L.T.); 2Institute of Analytical and Environmental Sciences, National Tsing Hua University, 101, Sec. 2, Kuang Fu Road, Hsinchu 30013, Taiwan; htviet1993@gmail.com

**Keywords:** supercapacitor, energy storage, coffee ground-derived biochar, graphene quantum dots (GQDs), erbium

## Abstract

A nanocomposite with erbium-doped graphene quantum dots embedded in highly porous coffee-ground-derived biochar (Er-GQD/HPB) was synthesized as a promising electrode material for a highly efficient supercapacitor. The HPB showed high porosity, with a large surface area of 1295 m^2^ g^−1^ and an average pore size of 2.8 nm. The 2–8-nanometer Er-GQD nanoparticles were uniformly decorated on the HPB, subsequently increasing its specific surface area and thermal stability. Furthermore, the intimate contact between the Er-GQDs and HPB significantly reduced the charge-transfer resistance and diffusion path, leading to the rapid migration of ions/electrons in the mesoporous channels of the HPB. By adding Er-GQDs, the specific capacitance was dramatically increased from 337 F g^−1^ for the pure HPB to 699 F g^−1^ for the Er-GQD/HPB at 1 A g^−1^. The Ragone plot of the Er-GQD/HPB exhibited an ultrahigh energy density of 94.5 Wh kg^−1^ and a power density of 1.3 kW kg^−1^ at 1 A g^−1^. Furthermore, the Er-GQD/HPB electrode displayed excellent cycling stability, and 81% of the initial capacitance remained after 5000 cycles. Our results provide further insights into a promising supercapacitance material that offers the benefits of both fast ion transport from highly porous carbons and electrocatalytic improvement due to the embedment of Er-doped GQDs to enhance energy density relative to conventional materials.

## 1. Introduction

The development of the modern economy and industry has manifestly increased the demand for healthcare monitoring and wearable and portable electronics worldwide. This demand requires paramount power supplies in energy storage devices [1]. The supercapacitor (SC) is an emerging electrochemical device, which plays a prominent role in energy storage systems because of its novel advantages, such as its light weight, flexibility, fast charge—discharge rate, and long lifespan [1,2]. Therefore, the exploitation of new and novel SC devices has become attractive because a vast amount of energy can be stored and delivered under quick output power density [1,3].

Various structured carbon materials, such as carbon nanotubes, activated carbon, the graphene family, and biochars have been used to provide an excellent performance of electric double layer capacitance (EDLC) due to their large surface area, good conductive nature, and available porosity for ion/electron transport [4,5,6]. The use of biomass-derived biochars as the EDLC material has recently received considerable attention [7,8,9,10,11]. Gao et al. synthesized hierarchically porous biochars for SC applications and found that 97% of the initial capacitance was preserved after 5000 cycles [7]. Another study used heteroatom-doped porous biochar to enhance the specific capacitance (C_s_), which reached up to 447 F g^−1^ at 0.2 A g^−1^ [8]. Although carbon-based materials have been widely studied as electrode materials for SC applications [7,8,9,10,11], improvements in their capacitive performance and electrical stability are still essential. More recently, the rare-earth (RE) elements, including La, Ce, Nd, Er, and Y have been considered as highly efficient dopants for increasing energy storage due to their potential to enhance electroactivity for rapid electron transfer to improve capacitive performance, as well as cycling stability [12,13]. A previous study used La-doped MnO_2_@GO nanocomposite as the electrode material and found that the La-MnO_2_@GO could deliver C_s_ of 729 F g^−1^ at 5 mV s^−1^ [14]. Dinari et al. [15] fabricated a Ce-doped NiCo-LDH@CNT nanocomposite for SC application, and 85.6% of the capacitance retention was obtained at 9000 cycles. These studies highlight the significance of RE-doped carbon-based nanocomposites for the improvement of the electrochemical performance of SC applications.

In addition to doping with RE elements, the combination of graphene quantum dot (GQD) with biochar is another effective method to improve the capacitive behavior of carbon-based electrodes for SC applications. GQDs can serve as surface modifiers to create a high number of edge states as the electrochemically active sites on the surface, resulting in the enhancement of the conductivity as well as the surface wettability, of electrode materials [16,17,18,19]. Rahimpour et al. [20] used ferrocenyl to modify a GQD/polypyrrole nanocomposite, and a C_s_ of 284 F g^−1^ at 2.5 A g^−1^ was achieved. Ganganboina et al. [21] recently fabricated a nano-sandwiched V_2_O_5_/GQD-based electrode material for SC systems. A good electrochemical performance with a capacitance of 572 F g^−1^ was obtained [21]. Several studies have also shown that doped GQDs, such as S-GQDs and N-GQDs, can create more active sites to improve the supercapacitive performance [22,23]. A previous study reported that the C_s_ of N-and-F-co-doped GQDs was 270 F g^−1^ at 1 mV s^−1^ [22]. When anchoring N-doped GQDs onto the rGO, the nanostructure showed an enhanced C_s_ of 416.5 F g^−1^ at 1 A g^−1^ [23], clearly demonstrating that doped GQDs can significantly improve supercapacitance. These results clearly demonstrate that the co-deposition of RE and GQDs onto biochar-based carbon materials forms a promising electrode material for SC application. However, few studies have focused on the synergistic effect of erbium species and GQDs on the electrochemical performances of SC applications.

Erbium (Er) and its derivatives have been reported as promising electroactive elements for advancing lithium-ion batteries with good cycling stability [24,25,26]. In addition, Er has multiple oxidation states, including Er^2+^ and Er^3+^, which can incorporate electroactive ErOOH to display a high C_s_ of 1811 F g^−1^ at 3 A g^−1^ under the pseudocapacitive behavior [26]. At the time of writing, few studies have combined Er elements with metal oxides to increase the interfacial ion and electron transport for reversible Faradaic reactions, and the doping of Er with GQDs (Er-GQDs) to enhance the electrochemical performance of biochars for highly efficient SC application has not yet been reported. Er-GQD nanocomposites are highly promising, as they can generate more electrochemically active sites and good hydrophilic properties for biochar-based materials to access electrolyte ions and electrons more readily.

In this study, an Er-doped GQD-deposited highly porous biochar (Er-GQD/HPB) composite was developed via the hydrothermal process as a novel electrode material for high-performance SC applications. As shown in Figure 1, waste coffee grounds (WCGs) were chosen as the raw materials for the fabrication of the HPB because the abundant N contents in WCGs can provide high conductivity and more active sites for electron transfer. Various amounts of Er at 1–20 mM were added onto the GQDs to form Er-GQDs, which were then hydrothermally deposited onto the HPB surface at 200 °C for 4 h. The scanning and transmission electron microscopic results (SEM/TEM) indicate that 2–8 nm Er-GQDs can be well dispersed onto HPB. The microstructure, surface property, and chemical species were further examined to elucidate the functional groups and thermal stability of the as-fabricated Er-GQD/HPB. The results of the analysis of the electrochemical properties, including cyclic voltammetry (CV), galvanostatic charge-discharge (GCD), and electrochemical impedance spectroscopy (EIS), clearly signify that the interaction between Er-GQDs and mesoporous HPB benefits fast ion transport, leading to an increase in power and energy density. Moreover, the composite’s cycling stability and the practical ability to illuminate a light-emitting diode (LED) were also investigated. The Ragone plot shows that the energy density was in the range of 56.4–94.5 Wh kg^−1^, corroborating the notion of the superiority of Er-GQD/HPB as the electrode material for SC applications.

## 2. Materials and Methods

### 2.1. Chemicals

Erbium nitrate pentahydrate (Er(NO_3_)_3_⋅5H_2_O), lactose, and potassium hydroxide were purchased from Sigma-Aldrich (Taufkirchen, Germany). 1-Methyl-2-pyrrolidone (NMP) was obtained from Merck Co (Frankfurter, Darmstadt, Germany). WCGs were collected from a local coffee shop (Seattle, Washington, US). All the other chemicals were of analytical grade. All the solutions were prepared by using deionized water (DI water, 18.2 MΩ cm) unless otherwise stated.

### 2.2. Synthesis of Er-GQD/HPB Nanocomposites

WCG powders were ground and sieved with a 100-mesh sieve and then boiled in ethanol for 2 h. The solutions were then soaked in hexane for 5 h and washed several times using DI water. Wet WCGs were dried at 60 °C for 12 h and placed in a tube furnace at 450 °C for 4 h at 10 °C min^−1^ under argon conditions. The final powders were crushed again homogeneously, and subsequently mixed with KOH at a WCGs:KOH ratio of 1:2 (*w*/*w*). Before activation, the mixture slurry was dried in a vacuum oven at 100 °C for 24 h and activated in a tube furnace at 850 °C for 2 h at 3.5 °C min^−1^ under Ar conditions. After cooling, the obtained HPB from WCGs was washed with 0.1 M HCl several times until the solution pH was close to neutral. The HPB was dried again and stored until use. The synthesis of Er-GQDs was prepared according to our previous work [27]. In brief, erbium-doped GQD solution (Er-GQD) was fabricated by adding various Er concentrations of 1, 5, 10, 15, and 20 mM into 5 mL of GQD solution. Next, 50 mg HPB was mixed with 5 mL Er-GQDs for 1 h and heated hydrothermally for 4 h at 200 °C to produce Er x-GQD/HPB nanocomposites, where x denotes the added concentration of Er (1–20).

### 2.3. Characterization

The surface morphologies of HPB-based materials were investigated using a SEM (Hitachi SU8010, Tokyo, Japan). The particle size of Er-GQDs was determined by a JEOL TEM (JEM-ARM 200F, Tokyo, Japan) and a high-resolution TEM (JEM-2010, Peabody, MA, USA). A Micrometrics ASAP 2020 analyzer (Norcross, GA, USA) was used to determine the surface areas and pore textures of HPB-based composites. Moreover, the crystallinity of HPB-based nanomaterials was recorded with an X-ray diffractometer (XRD, Bruker D8 Advanced, Bremen, Germany) equipped with Ni-filtered Cu Kα radiation (*λ* = 1.5405 Å) at 10–50° 2*θ*. The chemical compositions and surface functional groups of HPB-based nanomaterials were identified using an X-ray photoelectron spectrometer (XPS, Kanagawa, Japan) at 1486.6 ± 0.2 eV and Horiba Fourier transform infrared spectroscopy (FTIR, Minami-ku Kyoto, Japan), respectively. Raman spectra were recorded using a Bruker Senterra Raman microscope (Bremen, Germany), while the thermal stability was investigated from 30 to 800 °C at 10 °C min^−1^ in air using a Mettler Toledo thermogravimetric analyzer (DSC/TGA 3+ Star, Greifensee, Switzerland).

### 2.4. Electrochemical Analysis

The electrochemical properties of HPB-based nanomaterials were examined with an Autolab PGSTAT 302N (Metrohm Autolab B.V., Utrecht, The Netherlands) electrochemical workstation using 2 M KOH as the electrolyte. The working electrode was prepared by mixing Er-GQD/HPB and PVDF (90:10 wt/wt) in NMP. This solution was mixed homogeneously, placed onto carbon paper with an effective area of 1 × 1 cm^2^, and then dried overnight to obtain the Er-GQD/HPB electrode with mass loading of 1.3 mg cm^−2^. Moreover, Hg/HgO and platinum wire were used as the reference and counter electrodes, respectively. CV scan was performed at scan rates of 5–100 mV s^−1^ in the potential window of −1.2–0 V, while the GCD profile was obtained by varying the current densities from 1 to 30 A g^−1^. EIS spectra were performed in the frequency range of 0.1 Hz–100 kHz. A two-electrode Er-GQD/HPB||HPB system was prepared for SC application using 2 M KOH and porous cellulose membrane as the electrolyte and separator, respectively. Next, the electrode device was inserted between two glass slides and was fixed by the binder clips. For practical application, the asymmetric device of Er-GQD/HPB was used to illuminate a light-emitting diode (LED) at a current and forward voltage of 20 mA and 2 V, respectively.

## 3. Results and Discussion

### 3.1. Nanocomposite Characterization

The morphologies and nanostructures of the HPB-based nanomaterials were characterized first. Figure 1a illustrates the homogeneous distribution of highly porous structure of HBP. As shown in Figure 1b, the HPB surface was dominated by meso- and macro-pores after the chemical activation with KOH and carbonization, indicating the abundance of electroactive sites on its surface [28]. The HPB surface in Figure 1c became rough in comparison with that of the pure HPB (Figure 1a), and some small particles can be seen after the attachment of Er-GQD nanoparticles. The successful doping of Er-GQD on the HPB surface (yellow circles) is further verified by the TEM image (Figure 1d). In addition, the Er-GQDs particles, which had a mean size of 4.5 nm, were distributed uniformly on the HPB surface, with an interplanar spacing of 0.35 nm, which correlated with the (002) plane of the HPB (Figure 1e). The HRTEM image also confirms that the fringe lattice spacing of the Er-GQDs was 0.21 nm, corresponding to the (100) plane of the graphene [17,29]. The fast Fourier transform image derived from the HRTEM (Figure 1f and Appendix A) further signifies the hexagonal diffraction planes of the sp^2^ graphitic carbon (inset) in the HPB material and the presence of the Er-GQDs (Figure 1g and Appendix A).

Figure 2a shows the XRD patterns of HPB and Er 10-GQD/HPB nanocomposites synthesized from the WCGs. Two diffraction peaks at 2*θ* = 23° and 43° were observed, which were assigned as the (002) and (100) planes of the graphite and disordered carbon layer, respectively [8,30,31]. The appearance of the sharp peak at 23° 2*θ* signifies the regularity of the crystalline structure because the activation of the HPB at 850 °C resulted in a better layer alignment. After deposition with Er-GQDs, the peaks at 23° and 43° 2*θ* became broad, which was mainly due to the appearance of Er-GQDs on the surface of HPB. These results highlight the high-level crystallinity of the HPB and Er-GQD/HPB nanocomposites, which was in good agreement with the HRTEM image in Figure 1e. In addition, the Raman spectrum of the HPB in Figure 2b comprised two peaks at 1335 and 1563 cm^−1^, corroborating the presence of the D and G bands from the defect in the carbon structure and the vibrational bond involving the sp^2^ hybridized carbon atoms of hexagonal carbon structure, respectively [32,33]. After adding the Er-GQDs to the HPB, the peaks slightly shifted to 1329 and 1551 cm^−1^, respectively. Moreover, the *I*_D_/*I*_G_ ratio decreased from 1.0 for the pure HPB to 0.98 for the Er-GQD/HPB, which indicates that the addition of Er-GQD to the HPB might enlarge the sp^2^ structure to enhance the conductivity.

The FTIR spectrum of the HPB exhibited a broad OH peak centered at 3406 cm^−1^ (Figure 2c), depicting the hydrophilicity of the HPB-based materials. The peaks at 1232, 1320, and 1579 cm^−1^ are the three carbonaceous functional groups of C–O, C=C, and C=O, respectively. After the deposition of the Er-GQDs, the carbonaceous groups were still present in the FTIR spectrum of the Er-GQD/HPB, but the C–O and C=C bands red-shifted to 1085–1472 cm^−1^. Additionally, the peaks at 1320 and 520 cm^−1^ are the footprints of carboxylic groups of the Er-GQDs and the vibration Er-O bond, respectively, suggesting that the Er coordinated with the carboxylate in the GQD nanoparticles to form the Er-GQDs nanocomposite. Figure 2d displays the N_2_ adsorption–desorption curves of the HPB-based materials. The pure HPB exhibits a type IV isotherm with the H3 hysteresis loop in the P/P_0_ region of 0.45–0.95, indicating the mesoporous nature of the pure HPB. Initially, the HPB had a specific surface area of 1295 m^2^ g^−1^, with a pore size of 2.8 nm. After doping with Er-GQDs, the specific surface area of the Er-GQD/HPB increased to 1360 m^2^ g^−1^, while the pore size decreased slightly to 2.6 nm, highlighting that the Er-GQDs were embedded onto the surface of the HPB and, subsequently, increased the specific surface area, as well as decreasing the pore size of the Er-GQD/HPB nanocomposites.

The thermal stability of the HPB and Er-GQD/HPB was further studied. As illustrated in Figure 3a, two major weight-loss steps were observed for both the HPB and the Er-GQD/HPB from the TGA curves. In the first step of the TGA, a weight loss of 5% at 70–125 °C, coupled with a strong exothermic peak at 107 °C, was observed, which was mainly due to the evaporation of the H_2_O molecules in the HPB samples. For the Er-GQD/HPB, a weight loss of 8% occurred under 200 °C. The differential scanning calorimetry (DSC) peak indicates that the loss of the H_2_O molecules from the Er-GQD/HPB occurred at 130 °C (Appendix A). The second step involved the 25% and 13% weight losses in the HPB and Er-GQD/HPB, respectively, at 250–350 °C. Moreover, the weights of the HPB and Er-GQD/HPB decreased gradually at 318–540 °C due to the carbon burning. The weight loss was completed at 665 °C for the HPB and at 747 °C for the Er-GQD/HPB, and only 2% and 4% of the HPB and Er-GQD/HPB, respectively, were retained in air. These results indicate that the addition of Er-GQDs to HPB significantly increases the thermal stability of Er-GQD/HPB, resulting in a high overall degree of graphitization [34].

The surface chemistry and element species of the as-prepared HPB and Er-GQD/HPB nanocomposites were examined using XPS. As displayed in Figure 3b, the survey spectra of both the HPB and the Er-GQD/HPB exhibited C 1s, N 1s, and O 1s peaks centered at 285, 400, and 532 eV, respectively. Furthermore, the Er 4d peak at 169 eV was also observed in the Er-GQD/HPB, which was consistent with the reported result [27]. The deconvoluted spectrum of the Er-GQD/HPB exhibited a dominant C 1s peak at 289.3 eV, accompanied by three other major peaks at 288.4, 287.3, and 284.5 eV (Figure 3c), indicating the presence of the COOH, O=C–O, C=O, and C–O functional groups, respectively [27,35]. The last two C 1s peaks, located at 286.5 and 285.3 eV, indicated the dominant aliphatic/aromatic (sp^3^/sp^2^) carbons (e.g., C–H and C–C). These results suggest the presence of the original functional groups of GQDs [27,35]. Similarly, the deconvoluted O 1s spectrum shows the Er–O bonds at 531.8 and 530.9 eV (Figure 3d), which were assigned as the linkage between the Er and O elements in the Er-GQD/HPB [36]. In addition, the other two peaks, at 533.6 and 532.7 eV, are the carboxylic C=O and O−H groups, respectively. The N 1s core-level peak includes a major pyrrolic-N peak at 400.0 eV, a distinguishable quaternary-N at 401.6 eV, and a pyridinic peak at 399.1 eV (Figure 3e) [37]. It is noteworthy that the pyrrolic-N and pyridinic-N have electrochemically active functional groups, which can feature the pseudo-capacitive behavior of carbon materials [38]. Furthermore, the deconvolution of Er 4d shows two peaks, at 170.8 and 168.2 eV (Figure 3f), belonging to the 4d_3/2_ and 4d_5/2_ of the Er-O, respectively [27,39]. These results clearly indicate the successful doping of the Er-GQD onto the HPB, which resulted in an increase in surface area and enhanced thermal stability. Moreover, the meso- and macro-porous structure of the Er-GQD/HPB nanocomposite can create abundant electroactive sites on the surface to improve the electrochemical performance of nanocomposite.

### 3.2. Electrochemical Characterization of Er-GQD/HPB Nanocomposites

The electrochemical properties of the HPB-based nanocomposites were further characterized using the three-electrode system. Figure 4a displays the CV curves of the Er-GQD/HPB electrode materials at 100 mV s^−1^ in a voltage range of −1.2–0 V. All the CV curves showed a quasi-rectangular shape at a scan rate of 100 mV s^−1^, which slightly deviated from the ideal rectangular shape of EDLC. This indicated the occurrence of quickly reversible faradaic reactions and superior charge storage for the Er-GQD/HPB nanocomposite at Er loadings of 1–20 mM [40,41]. Figure 4b shows the C_s_ of the Er-GQD/HPB at different Er loadings as a function of the scan rate. The added amount of Er-GQD had a different impact on the capacitance of Er-GQD/HPB. It is clear that the C_s_ increased from 282 F g^−1^ for the Er 1-GQD/HPB to 603 F g^−1^ for the Er 10-GQD/HPB at 5 mV s^−1^. However, the capacitance of the Er-GQD/HPB decreased as the loading of the Er increased to 15–20 mM, and the C_s_ values decreased to 468 and 417 F g^−1^ for the Er 15-GQD/HPB and Er 20-GQD/HPB, respectively, which suggests that the loading content of the Er at 10 mM is optimal to provide superior C_s_. In addition, the increase in the scan rate resulted in a decrease in C_s_ for all the Er-GQD/HPB electrode materials, and the Er 10-GQD/HPB still retained 47% of its initial capacitance at 100 mV s^−1^. Appendix A shows the CV curves of the HPB and the pure graphite paper. Similar to the Er-GQD/HPB nanomaterial, a quasi-rectangular-shaped CV curve was observed at 100 mV s^−1^. However, the C_s_ values were only 195 and 210 F g^−1^, suggesting that the addition of Er-GQD obviously enhanced the C_s_ of the Er x-GQD/HPB materials.

Figure 4c shows the GCD curves of the Er x-GQD/HPB at 1 A g^−1^. Similar to the CV curves, the GCD curves of the Er-GQD/HPB electrode materials showed a nearly triangle-shaped curve during the charge and discharge processes, which evidently demonstrates the reversible redox mechanism related to the intercalation/deintercalation of electrolyte ions onto Er-GQD/HPB [40,42]. It is worth noting that the longest discharge time in the series of Er-GQD/HPB electrodes was found at 1400 s for the GCD curve of the Er 10-GQD/HPB, indicating that the enhanced EDLC behavior provided the superior C_s_ of this electrode material. Figure 4d displays the C_s_ of the Er-GQD/HPB electrode materials, varying from 1 to 10 A g^−1^. The capacitance was ranked in the following order: Er 10-GQD/HPB > Er 15-GQD/HPB > Er 20-GQD/HPB > Er 5-GQD/HPB > Er 1-GQD/HPB. This tendency was highly correlated with the CV results and signified that Er 10-GQD loading was sufficient to achieve the optimal capacitive performance of the Er-GQD/HPB nanocomposites.

In this study, an optimal loading of Er-GQD onto HPB for enhanced electrochemical performance was observed. Low Er loading (i.e., Er 1-GQD/HPB and Er 5-GQD/HPB) resulted in the poor incorporation of Er-GQD and the carbon backbone of HPB, causing an inactive contact between active sites in supercapacitive reactions. By contrast, the overloading of dopants was likely to lead to the rapid aggregation of Er-GQD in the carbon pore, preventing the transport of electrons/ions [17,43]. The superior capacitive performance of the Er 10-GQD/HPB was likely due to its large surface area and excellent thermal stability. As shown in Appendix A, the Er 10-GQD/HPB was more thermally stable than the other Er-GQD/HPB, with a residual mass of 4% after 700 °C. Ganganboina et al. [21] embedded GQDs in V_2_O_5_ nanosheets and found that the superior stability of capacitive materials is a unique property that can keep electrode structures stable, and subsequently enhances ion penetration and cycling life. Hu et al. [43] also reported that the weight loss and the structural collapse during the insertion/deintercalation of ions is the main reason for the decrease in the C_s_ of the electrode materials. These results explicitly demonstrate that the high surface area and thermal stability of Er-GQD/HPB electrode materials can keep their structure stable, resulting in the production of high capacitance.

EIS tests were further carried out to examine the conductivity of electrode materials and their mechanism of charge transport, as well as the ion migration rate at the electrode/electrolyte interface (Appendix A). The Nyquist plots represent a traditional EIS curve, consisting of a semicircle and a straight line in the high- and low-frequency regions, respectively. It is noted that the intercept of the semicircle on the real Z’ axis is the equivalent series resistance (R_s_), while the diameter of the semicircle indicates the charge-transfer resistance (R_ct_), and the straight line corresponds to the ion diffusion resistance [44]. It is clear that the Er 10-GQD/HPB showed the lowest R_s_ value, of 1.7 Ω, which indicates the increase in its conductivity after the appropriate loading of the Er-GQD, in comparison with the R_s_ value of 5 Ω in the pure HPB (Appendix A). Similarly, the R_ct_ of the Er 10-GQD/HPB showed the smallest value of 3.9 Ω.

To further understand the capacitive behavior of the Er-GQD/HPB electrode materials, the influence of the scan rate and current density on the Er 10-GQD/HPB was examined explicitly. As illustrated in Figure 5a, a nearly rectangular-shaped curve was observed, and the curve started to deviate from the ideally rectangular shape when the scan rate increased. It is noted that the quasi-rectangular shape of the CV curve remained when the scan rate reaches 100 mV s^−1^, which was a typical characteristic of excellent electrochemical reversibility and outstanding capacitive behavior. At the same time, the GCD curves of the Er 10-GQD/HPB electrode exhibited a relatively good and symmetrical regular triangular shape (Figure 5b). The introduction of a long discharging time compared to the charging time represented the fast reversible responses [42]. Appendix A shows the change in the CV and GCD curves of the pure HPB at different scan rates and current densities. Although the CV curves of the HPB exhibited the ideal rectangular shape of the EDLC (Appendix A), the derived C_s_ was smaller than that of the Er 10-GQD/HPB at all the tested scan rates. In addition, the charging time of the Er 10-GQD/HPB, accounting for 1400 s at 1 A g^−1^, was much longer than that of the pure HBP (550 s) (Appendix A). It is noteworthy that the C_s_ values of the pure HPB were in the range of 313–195 F g^−1^ at 5–100 mV s^−1^ and 337–210 F g^−1^ at 1–30 A g^−1^ (Figure 5c), and that the addition of the Er-GQD significantly enhanced the electrochemical performance of the HPB-based nanomaterials. As shown in Figure 5d, the Er 10-GQD/HPB electrode material displayed a very competitive C_s_ of 699 F g^−1^ at 1 A g^−1^ and remained at 338 F g^−1^ at 30 A g^−1^. Similar to the GCD results, the Er 10-GQD/HPB electrode had an excellent C_s_ of 603 F g^−1^ at 5 mV s^−1^ and remained at 282 F g^−1^ at 100 mV s^−1^, signifying the superiority of the developed Er-GQD/HPB electrode materials for SC applications.

The diffusion and adsorption of ion/electrons at the electrode surface are generally limited at high scan rates. In this study, the Er 10-GQD/HPB exhibited a relatively high C_s_ at a high scan rate and current density, presumably due to the fact that the Er 10-GQD/HPB consisted in a large number of edged active sites of the Er-GQD, as well as the highly micro-mesoporous structure of the HPB support. Moreover, the ultrahigh electrochemical performance of the Er 10-GQD/HPB was likely due to the extraordinary collaboration between the porous structure of the HPB and the unique configuration of the Er-GQD, resulting in the reduction in charge transfer resistance, as well as the enhancement of the structural stability, of the nanocomposite. The short straight line in the Nyquist plot of the Er 10-GQD/HPB confirms its rapid ion migration rate, which resulted from the beneficially low resistance due to the mesoporous structure of the HPB. Moreover, the addition of the Er-GQDs provided electrochemically active sites to store the charges and access ions. As a result, the novel Er-GQD/HPB electrode material can benefit from these characteristics, and the optimal Er 10-GQD/HPB material exhibits ideally capacitive behavior with high conductivity and enhanced capacitance, which can be used for asymmetric SC applications.

Table 1 compares the electrochemical performances of the Er 10-GQD/HPB electrode materials with the reported results by using carbon nanomaterials as the electrodes. It is clear that the C_s_ of the most strongly GQD-based materials are in the range of 284–645 F g^−1^. Xing et al. [18] used N-GQD/PANI as the electrode material, and C_s_ of 506 F g^−1^ was obtained at 0.5 A g^−1^ in the presence of 1 M H_2_SO_4_. Moreover, Nirmaladevi et al. [10] combined MnO_2_ with biochar (BC@MnO_2_) as the effective electrode materials. The C_s_ of the BC@MnO_2_ reached 512 F g^−1^ at 0.5 A g^−1^ when 1 M Na_2_SO_4_ was used as the electrolyte. In this study, the optimized Er 10-GQD/HPB nanocomposite achieved a superior C_s_ of 699 F g^−1^ at 1 A g^−1^, which was higher than those mentioned above. The Er-GQD nanocomposite is thus a potential dopant with which to improve the electrical properties of biochar-based materials.

### 3.3. Electrochemical Performance of Asymmetric Supercapacitor

To demonstrate the capacitive behavior of the electrode composite in a practical application, an Er-GQD/HPB-based supercapacitive device was fabricated by using Er 10-GQD/HPB and HPB nanocomposites as the positive and negative electrodes, respectively. In particular, the optimal voltage window of Er 10-GQD/HPB||HPB at 100 mV s^−1^ is illustrated in Appendix A, and the voltage of the Er 10-GQD/HPB||HPB cell reached 2.5 V. The electrochemical performance was further evaluated via the CV and GCD tests. The CV curves of the asymmetric device clearly show the ideal rectangular shape, with no shape distortion at scan rates of 5–100 mV s^−1^ (Figure 6a). The GCD curves in Figure 6b consistently displayed triangular shapes at 1–12 A g^−1^. In addition, the IR drop was indistinguishable in all the GCD curves, suggesting the excellent capacitive behavior of this SC.

The good charge-transfer kinetics and ion diffusion of the asymmetric supercapacitor Er-GQD/HPB were further clarified by EIS measurement. As presented in Figure 6c, the Nyquist plot with the insertion of the equivalent circuit model (Randles circuit) of the Er-GQD/HPB supercapacitive device. In addition, the equivalent circuit with their fitted parameters were shown in Table 2. It is clear that the Er-GQD/HPB showed a small R_s_ of 1.5 Ω, and a short semicircle R_ct_ of 1.2 Ω. Moreover, a short vertical straight line was observed in comparison with the Er 10-GQD/HPB. These results confirm the prominent structure and electrochemical features of the active Er 10-GQD/HPB material, which can be applied to construct excellent SC cells.

The Ragone plots express the relationship between the energy and power densities of the Er-GQD/HPB. As plotted in Figure 6d, the device showed an energy density of 94.5 Wh kg^−1^, with a corresponding power density of 1.3 kW kg^−1^ at 1 A g^−1^. Correspondingly, it preserved 60% of its energy density (56.4 kW kg^−1^) as the power density increased to 15 kW kg^−1^. Mane et al. [14] used 3%La–MnO_2_@GO fabricated symmetric SC, and the energy and power densities were in the range of 64−10 Wh kg^−1^ and 1–1.5 kW kg^−1^, respectively. When Sangabathula et al. [45] combined GQDs with a Mo-doped nickel sulfide (MNS-G) nanocomposite as the electrode material for an asymmetric supercapacitor, the energy density was 38.9–11.1 Wh kg^−1^ and the power density of 0.4–6.7 kW kg^−1^. Lei et al. [8] reported that the energy and power densities of heteroatom-doped porous-biochar-based SC were in the range of 12.8–6 Wh kg^−1^ and 0.05–5 kW kg^−1^, respectively. These results explicitly signify that the Er 10-GQD/HPB||HPB electrode material is a superior electroactive material to the symmetric HPB||HPB and to the various, previously reported, heteroatom-doped metal oxide/porous biochar nanocomposites and GQD nanomaterials [8,14,43,45,46,47].

The cycling stability of the Er 10-GQD/HPB||HPB device is also an important characteristic of its application. As illustrated in Figure 6e, the asymmetric SC device exhibited a superior long-standing stability for up to 5000 cycles, and the capacitance retained 81% at 1 A g^−1^. Moreover, charge–discharge curves between the first and last cycles were similar, exhibiting the excellent electrochemical stability of the device during the charge–discharge processes. This signifies the contribution of the unique, highly porous biochar and electroactive characteristics of the Er-GQD, as well as the intimate interface between the electrolyte and electrode material. Moreover, two asymmetric Er 10-GQD/HPB||HPB SC devices were connected to turn on a green LED (2 V) to demonstrate the real application of the SC. As displayed in Appendix A, the as-developed SC could easily keep the LED lamp lighting for more than 53 s after charging for only 5 s.

## 4. Conclusions

In this study, the Er-GQD-embedded highly porous coffee-waste-biochar nanocomposite was successfully developed for a highly efficient SC application. The specific surface area of the Er 10-GQD/HPB nanocomposite can be up to 1365 m^2^ g^−1^, with excellent pore size distribution. The amount of added Er has a significant impact on the electrochemical performance of the Er-GQD/HPB, and the C_s_ increases from 172 F g^−1^ at 1 mM Er to 699 F g^−1^ at 10 mM Er, and then decreases to 562 F g^−1^ as the Er concentration increases to 15–20 mM at 1 A g^−1^. The as-prepared asymmetric Er 10-GQD/HPB||HPB SC device exhibits a superior electrochemical performance, with an ultrahigh energy density of 94.5 Wh kg^−1^ at a power density of 1.3 kW kg^−1^. Moreover, the energy density can remain at 56.4 Wh kg^−1^ when the power density is 15 kW kg^−1^. This can be ascribed to the high abundance of active sites in and electroactive activity of the Er-GQD, which enhances the electronic/ionic-transfer pathways for high energy storage. Furthermore, the as prepared Er-GQD/HPB benefits from the high conductivity of the mesoporous HPB to reduce the charge-transfer resistance as well as to increase the energy density. The incorporation of the Er-GQD into the HPB also enhances the overall stability of the device, which can exhibit excellent cycling stability after 5000 cycles at 2 M KOH. The results of this study provide a promising strategy to develop a platform by using agricultural wastes as the raw materials for the fabrication of highly conductive carbon materials with large specific surface areas, interconnected porous channels, and low impedance. Moreover, the doping of RE elements is another possible alternative to enhance the electrochemical performance of porous carbon materials for energy storage applications.

## Data Availability

Not applicable.

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
