# Peer review of "Erbium-Doped GQD-Embedded Coffee-Ground-Derived Porous Biochar for Highly Efficient Asymmetric Supercapacitor"

_nanomaterials, 2022, doi:10.3390/nano12111939_

Round 1
Reviewer 1 Report
The current manuscript demonstrates the supercapacitor application of Er-doped graphene quantum dot embedded biochar. The manuscript has significant data to be accepted for publication in this journal. Therefore, I recommend acceptance of this manuscript after minor revision.
My comments are given below -
- A comparative table of the electrochemical performance of the GQD-based biochar electrode should be given with other related electrodes.
- The electrochemical perform should also be expressed in specific capacity with the unit of C/g or mAh/g.
- The equivalent circuit used for the fitting of the Nyquist plot should be provided.
- The mass loading of the electrode materials should be provided in the manuscript.
Author Response
Reply to comments of reviewer #1
Comment #1: A comparative table of the electrochemical performance of the GQD-based biochar electrode should be given with other related electrodes.
Reply: A supporting table to compare the electrochemical performance of our material and other related materials reported in the literature has been added to the revised manuscript (line 372-385).
Table 1. compares the electrochemical performance of the Er 10-GQD/HPB electrode materials with those literally reported results by using carbon nanomaterials as the electrode. It is clear that the specific capacitance of most GQD-based are in the range of 284-645 F g-1. Xing et al. [18] has used N-GQD/PANI as the electrode material, and the Cs of 506 F g-1 was obtained at 0.5 A g-1 in the presence of 1 M H2SO4. Moreover, Nirmaladevi et al. [10] has combined MnO2 with biochar as the effective electrode materials. The specific capacitance of BC@MnO2 could be up to 512 F g-1 at 0.5 A g-1 when 1 M Na2SO4 was used as the electrolyte. In this study, the optimized Er 10-GQD/HPB nanocomposite achieves the superior Cs of 699 F g-1 at 1 A g-1, which is relatively higher than those mentioned above. The Er-GQDs nanocomposite thus becomes a potential dopant to improve the electrical properties of biochar-based material.
Table 1. Comparative electrochemical performance of the GQD-based biochar electrode with other related electrode materials.
|
Electrode materials |
Electrolyte |
Potential (V) |
Specific capacitance (F g-1) |
Current density (A g-1) |
Ref. |
|
GQD-Fc/PPy |
1 M Na2SO4 |
-0.2 to 1 |
284 |
2.5 |
[1] |
|
NGQDs/PANI |
1 M H2SO4 |
-0.2 to 0.8 |
506 |
0.5 |
[2] |
|
3DMoS2-N-GQDs-rGO |
1 M Na2SO4 |
-0.8 to 0.2 |
416.5 |
1 |
[3] |
|
S,N-GQDP2 |
1 M H2SO4 |
-0.4 to 0.6 |
645 |
0.5 |
[4] |
|
N-doped AC |
6 M KOH |
-1 to 0 |
382.6 |
0.5 |
[5] |
|
HMC-24 |
1 M H2SO4 |
-0.4 to 0.6 |
447 |
0.2 |
[6] |
|
BC@MnO2 |
1 M Na2SO4 |
-0.2 to 1 |
512 |
0.5 |
[7] |
|
Er-GQD/HPB |
2 M KOH |
-1.2 to 0 |
699 |
1 |
This work |
Comment #2: The electrochemical perform should also be expressed in specific capacity with the unit of C/g or mAh/g.
Reply: We thank the reviewer’s suggestion. Usually the C/g or mAh/g is widely used in battery related research. In this study, we mainly focus on the use of Er-GQD/HPB material for supercapacitor application. Therefore, the unit of “F/g” is more suitable in this study, which can be compared to the other reported supercapacitor studies.
Comment #3: The equivalent circuit used for the fitting of the Nyquist plot should be provided.
Reply: The equivalent circuit, which is derived from the fitting of the Nyquist plot, has been added to the revised manuscript.
Line 397-400: The good charge transfer kinetic and ion diffusion of asymmetric supercapacitor Er-GQD/HPB were further clarified by EIS measurement. Figure 6c presents the Nyquist plot with equivalent circuit model (Randles circuit) of Er-GQD/HPB supercapacitive device. In addition, the equivalent circuit with their fitted parameters were shown in a Table 2. It is clear that the Er-GQD/HPB shows a small Rs of 1.5 Ω, and a short semicircle Rct of 1.2 Ω. Moreover, a short vertical straight line is observed in comparison with Er 10-GQD/HPB. These results confirm the prominent structure and electrochemical features of the active Er 10-GQD/HPB material, which can be applied to construct excellent supercapacitor cells.
Line 425-426: Table 2. Fitting parameters of the equivalent circuit.
|
Element |
Value |
|
Rs |
1.5 |
|
Rct |
1.2 |
|
Wo1 |
0.2 |
|
CPE1 |
0.5 |
|
CPE2 |
0.9 |
Comment #4 The mass loading of the electrode materials should be provided in the manuscript.
Reply: The mass loading of Er-GQD/HPB on the electrode material was 1.3 mg. We have included this information in the revised manuscript.
Line 154-155: The working electrode was prepared by mixing Er-GQD/HPB and PVDF (90:10 wt/wt) in NMP. This solution was mixed homogeneously and placed onto a carbon paper with an effective area of 1´1 cm2 and then dried overnight to obtain the Er-GQD/HPB electrode with the mass loading of 1.3 mg cm-2.

Reviewer 2 Report
This is a carefully performed high-quality work, reporting supercapacitors using waste coffee ground. In Abstract the acronyms GQD and HPB should be (more clearly) explained. Frequency information in the impedance Nquist plots should be indicated. Descriptions like 'Short straight line confirms rapid migration' seems not sufficient. More detailed EIS for supercapacitors are being performed. Please provide more quantitative EIS analysis to support related discussions.
Author Response
Reply to the comments of Reviewer #2
Comment #1: In Abstract the acronyms GQD and HPB should be (more clearly) explained.
Reply: We have added the full name of GQD and HPB in the abstract.
Line 13: The Erbium-doped graphene quantum dots embedded highly porous coffee ground derived biochar (Er-GQD/HPB) nanocomposite is synthesized as promising electrode material for highly efficient supercapacitor application.
Comment #2 Frequency information in the impedance Nyquist plots should be indicated.
Reply: We have included the frequency information to the revised manuscript.
Line 158-159: Moreover, Hg/HgO and platinum wire were used as the reference and counter electrodes, respectively. CV scan was performed at scan rates of 5–100 mV s-1 in the potential window of -1.2–0 V, while the GCD profile was obtained by varying the current densities from 1 to 30 A g-1. EIS spectra were performed in the frequency range of 0.1–100 kHz.
Comment #3: Descriptions like 'Short straight line confirms rapid migration' seems not sufficient.
Reply: We have included the extra information to the revised manuscript.
Line 361-363: Moreover, the ultrahigh electrochemical performance of Er 10-GQD/HPB is likely due to the extraordinary collaboration between the porous structure of HPB and the unique configuration of Er-GQD, resulting in the reduction in charge transfer resistance as well as the enhancement of structural stability of nanocomposite. The short straight line in Nyquist plot of Er 10-GQD/HPB confirms its rapid ion migration rate, which results from the benefit of a low resistance due to the mesoporous structure of HPB.
Comment #4 More detailed EIS for supercapacitors are being performed. Please provide more quantitative EIS analysis to support related discussions.
Reply: We have added more information of EIS analysis including the equivalent circuit to the revised manuscript.
Line 397 – 400: The good charge transfer kinetic and ion diffusion of asymmetric supercapacitor Er-GQD/HPB were further clarified by EIS measurement. Figure 6c presents the Nyquist plot with equivalent circuit model (Randles circuit) of Er-GQD/HPB supercapacitive device. In addition, the equivalent circuit with their fitted parameters were shown in a Table 2. It is clear that the Er-GQD/HPB shows a small Rs of 1.5 Ω, and a short semicircle Rct of 1.2 Ω. Moreover, a short vertical straight line is observed in comparison with Er 10-GQD/HPB. These results confirm the prominent structure and electrochemical features of the active Er 10-GQD/HPB material, which can be applied to construct excellent supercapacitor cells.
Line 425-426: Table 2. Fitting parameters of the equivalent circuit.
|
Element |
Value |
|
Rs |
1.5 |
|
Rct |
1.2 |
|
Wo1 |
0.2 |
|
CPE1 |
0.5 |
|
CPE2 |
0.9 |

Reviewer 3 Report
1. In your paper, please highlight the following:
why this work has been done,
what is new about it,
highlight the novelty (has such work been done before, if so, why you are doing this work)
2. Many space errors/punctuation errors must be solved. Abbreviations must be clearly followed throughout the manuscript. Parameters are defined several times in the manuscript.
3. The introduction section is poorly constructed. Author needs to elaborate on the background of the research in detailed manner.
4. The Authors are encouraged to review the form and the English of the manuscript.
5. After discussing each section, it is recommended to provide a short conclusion.
6. Nyquist plot must be discussed with equivalent circuit and their fitted parameters in tables.
7. In order to verify the validity of the electrochemical performance of the marteirals, authors need to compare their results with reported experimental data in tables.
8. Conclusion section- must focus on future directions? General statements need to be removed. Authors are suggested to be more specific in their conclusions.
Author Response
Reply to comments of Reviewer #3
Comment #1: In your paper, please highlight the following: why this work has been done,
what is new about it, highlight the novelty (has such work been done before, if so, why you are doing this work)
Reply: The authors thank the reviewer’s comment and suggestion. The literature survey of previously reported studies as well as the novelty of this study has been added to the revised manuscript.
Line 49-89: Another study has used heteroatom-doped porous biochar to enhance the specific capacitance (Cs), and the Cs was up to 447 F g-1 at 0.2 A g-1 [8]. Although carbon-based materials have been widely studied as electrode materials for SC applications [7-11], the improvement of their capacitive performance and electrical stability is still essential. More recently, the rare-earth (RE) elements including La, Ce, Nd, Er, and Y are considered as a highly efficient dopant for advancing energy storage due to their promising potential to enhance the electroactivity for rapid electrons transfer to improve the capacitive performance as well as the cycling stability [12, 13]. A previous study has used La-doped MnO2@GO nanocomposite as the electrode material and found that the La-MnO2@GO could deliver the Cs of 729 F g-1 at 5 mV s-1 [14]. Dinari et al. [15] fabricated the Ce-doped NiCo-LDH@CNT nanocomposite for SC application, and 85.6% of capacitance retention was obtained after 9000 cycles. These studies highlight the significance of RE-doped carbon-based nanocomposites for the improvement of the electrochemical performance on SC applications.
In addition to the doping of RE elements, the combination of graphene quantum dot (GQD) with biochar is another effective path to improve the capacitive behavior of carbon-based electrodes for SC application. GQDs can serve as the surface modifier to create high numbers of edge state as the electrochemically active sites on the surface, resulting in the enhancement of conductivity as well as surface wettability of the electrode materials [16-19]. Rahimpour et al. [16] have used ferrocenyl to modify GQD/polypyrrole nanocomposite and a Cs of 284 F g-1 at 2.5 A g-1 was achieved. Ganganboina et al. [21] have recently fabricated the nano-sandwiched V2O5/GQD based electrode material for SC system. A good electrochemical performance with a capacitance of 572 F g-1 was obtained [21]. Several studies have also depicted that the doped GQDs like S-GQDs and N-GQDs can create more active sites to improve the supercapacitive performance [22, 23]. A previous study has reported that the Cs of N, F-codoped GQDs was 270 F g-1 at 1 mV s-1 [22]. When anchoring N-doped GQDs onto rGO, the nanostructure showed an enhanced Cs of 416.5 F g-1 at 1 A g-1 [23], clearly depicting that the doped GQDs can significantly improve the supercapacitance. These results clearly elaborate that the co-deposition of RE and GQDs onto biochar-based carbon material would be a promising electrode material for SC application. However, only limited studies have focused on the synergistic effect of erbium species and GODs on electrochemical performance of SC applications.
Erbium (Er) and its derivatives have been reported as promising electroactive elements for advancing lithium-ion batteries with good cycling stability [24-26]. In addition, Er has multiple oxidation states, including Er2+ and Er3+, which can incorporate electroactive ErOOH to display a high Cs of 1811 F g-1 at 3 A g-1 under the pseudocapacitive behavior [26]. Up to now, only limited studies have combined Er element with metal oxides to increase the interfacial ion and electron transport for reversible Faradaic reactions, and the doping of Er with GQDs (Er-GQDs) to enhance the electrochemical performance of biochars for highly efficient SC application has not been reported yet. The Er-GQDs nanocomposite is highly promising as it can generate more electrochemically active sites and good hydrophilic properties for biochar-based material to access electrolyte ions and electrons more readily.
Comment #2: Many space errors/punctuation errors must be solved. Abbreviations must be clearly followed throughout the manuscript. Parameters are defined several times in the manuscript.
Reply: We thank the reviewer for the suggestion. The manuscript has been spell-checked and the manuscript have been read and corrected by the colleague who is the native English speaker. Moreover, all the space/punctuation errors have been corrected.
Comment #3. The introduction section is poorly constructed. Author needs to elaborate on the background of the research in detailed manner.
Reply: The introduction section has been re-organized and several sentences have been re-written. Please see the details in the Reply to comment #1.
Comment #4. The Authors are encouraged to review the form and the English of the manuscript.
Reply: We thank the reviewer for the suggestion. The manuscript has been spell-checked and the manuscript have been read and corrected by the colleague who is the native English speaker. We believe the manuscript quality has been improved and is now more readable.
Comment #5. After discussing each section, it is recommended to provide a short conclusion.
Reply: The short conclusion has been added to each section in the revised manuscript.
Line 261-265: Furthermore, the deconvolution of Er 4d shows two peak at 170.8 and 168.2 eV (Figure 3f), belonging to the 4d3/2 and 4d5/2 of Er-O, respectively [23, 34]. These results clearly indicate the successful doping of Er-GQD onto HPB, and subsequently results in the increase in surface area and the enhanced thermal stability. Moreover, the meso- and macro-porous structure of Er-GQD/HPB nanocomposite can create abundant electroactive sites on the surface to improve the electrochemical performance of nanocomposite.
Line 351-352: Similar to the GCD results, the Er 10-GQD/HPB electrode has a great Cs of 603 F g-1 at 5 mV s-1 and remains 282 F g-1 at 100 mV s-1, signifying the superiority of the developed Er-GQD/HPB electrode materials for SC applicaitons.
Line 365-367: As a result, the novel Er-GQD/HPB electrode material can gain benefits from these characteristics, and the optimal Er 10-GQD/HPB material exhibits the ideally capacitive behavior with high conductivity and enhanced capacitance, which can be used for asymmetric SC application.
Comment #6. Nyquist plot must be discussed with equivalent circuit and their fitted parameters in tables.
Reply: The equivalent circuit, which is derived from the fitting of Nyquist plot, has been added to the revised manuscript.
Line 397 – 400: The good charge transfer kinetic and ion diffusion of asymmetric supercapacitor Er-GQD/HPB were further clarified by EIS measurement. Figure 6c presents the Nyquist plot with equivalent circuit model (Randles circuit) of Er-GQD/HPB supercapacitive device. In addition, the equivalent circuit with their fitted parameters were shown in a Table 2. It is clear that the Er-GQD/HPB shows a small Rs of 1.5 Ω, and a short semicircle Rct of 1.2 Ω. Moreover, a short vertical straight line is observed in comparison with Er 10-GQD/HPB. These results confirm the prominent structure and electrochemical features of the active Er 10-GQD/HPB material, which can be applied to construct excellent supercapacitor cells.
Line 425-426: Table 2. Fitting parameters of the equivalent circuit.
|
Element |
Value |
|
Rs |
1.5 |
|
Rct |
1.2 |
|
Wo1 |
0.2 |
|
CPE1 |
0.5 |
|
CPE2 |
0.9 |
Comment #7. In order to verify the validity of the electrochemical performance of the materials, authors need to compare their results with reported experimental data in tables.
Reply: A supporting table to compare the electrochemical performance of our material and other related materials reported in the literature has been added to the revised manuscript (line 372-385).
Table 1. Comparative electrochemical performance of the GQD-based biochar electrode with other related electrode materials.
|
Electrode materials |
Electrolyte |
Potential (V) |
Specific capacitance (F g-1) |
Current density (A g-1) |
Ref. |
|
GQD-Fc/PPy |
1 M Na2SO4 |
-0.2 to 1 |
284 |
2.5 |
[1] |
|
NGQDs/PANI |
1 M H2SO4 |
-0.2 to 0.8 |
506 |
0.5 |
[2] |
|
3DMoS2-N-GQDs-rGO |
1 M Na2SO4 |
-0.8 to 0.2 |
416.5 |
1 |
[3] |
|
S,N-GQDP2 |
1 M H2SO4 |
-0.4 to 0.6 |
645 |
0.5 |
[4] |
|
N-doped AC |
6 M KOH |
-1 to 0 |
382.6 |
0.5 |
[5] |
|
HMC-24 |
1 M H2SO4 |
-0.4 to 0.6 |
447 |
0.2 |
[6] |
|
BC@MnO2 |
1 M Na2SO4 |
-0.2 to 1 |
512 |
0.5 |
[7] |
|
Er-GQD/HPB |
2 M KOH |
-1.2 to 0 |
699 |
1 |
This work |
Comment #8. Conclusion section- must focus on future directions? General statements need to be removed. Authors are suggested to be more specific in their conclusions.
Reply: The authors thank the reviewer’s comment. We have highlighted the significance and future direction of our study in the conclusion.
Line 454-459: The incorporation of Er-GQD to HPB also enhances the overall stability of the device, which can exhibit excellent cycling stability after 5000 cycles in 2 M KOH. Results of this study provide a promising strategy to develop a platform by using agricultural wastes as the raw materials for the fabrication of highly conductive carbon materials with large specific surface area, interconnected porous channel, and low impedance. Moreover, the doping of RE elements is another possible alternative to enhance the electrochemical performance of porous carbon materials for energy storage application.
